# Understanding Diversity of Policies, Functionalities, and Operationalization of Immunization Information Systems and Their Impact: A Targeted Review of the Literature

**DOI:** 10.3390/vaccines11071242

**Published:** 2023-07-15

**Authors:** Elizabeth A. Donckels, Luke Cunniff, Nina Regenold, Kaitlyn Esselman, Erik Muther, Alexandra Bhatti, Amanda L. Eiden

**Affiliations:** 1Real Chemistry Market Access, San Francisco, CA 94108, USAnregenold@realchemistry.com (N.R.); kesselman@realchemistry.com (K.E.); emuther@realchemistry.com (E.M.); 2Merck & Co., Inc., Rahway, NJ 07065, USA; luke.cunniff@merck.com (L.C.); amanda.eiden@merck.com (A.L.E.)

**Keywords:** vaccine, vaccine policy, health law, immunization, immunization information systems, digital health

## Abstract

The COVID-19 pandemic has focused attention on the use of immunization information systems (IIS) to record and consolidate immunization records from a variety of sources to generate comprehensive patient immunization histories. Operationalization of IIS in the United States is decentralized, and as such, there are over 60 different IIS with wide variations in enabling policies and functionalities. As such, the policies that inform the development and operation of those sub-national IIS exist at the state and sometimes city levels. A targeted literature review was conducted to identify IIS policies and functionalities and assess their impact. The authors identified articles published from 2012 to 2022 that discussed or evaluated IIS policies and functionalities and screened titles, abstracts, and full text for inclusion. When selected for inclusion, authors extracted IIS policy/functionality characteristics and qualitative or quantitative outcomes of their implementation, where applicable. The search terms yielded 86 articles, of which 39 were included in the analysis. The articles were heterogeneous with respect to study design, interventions, outcomes, and effect measures. Out of the 17 IIS policies and functional components identified in the targeted literature review, the most commonly evaluated were provider-based patient reminder/recall, IIS-based centralized reminder/recall, and clinical decision support. Patient reminder/recall had the most published research and was associated with increased vaccination rates and vaccine knowledge. Despite the lack of quantitative evidence, there is a consensus that immunization data interoperability is critical to supporting IIS data quality, access, and exchange. Significant evidence gaps remain about the effectiveness of IIS functionalities and policies. Future research should evaluate the impact of policies and functionalities to guide improved utilization of IIS, increase national interoperability and standardization, and ultimately improve vaccination coverage and population health.

## 1. Introduction

Immunization information systems (IIS) are confidential databases designed to record and consolidate vaccine doses administered by participating providers within a geographic area, to create comprehensive patient immunization records [1]. There are over 60 IIS in the United States (US) operating at the sub-national level in all fifty states, Washington, D.C., and select cities and territories [2]. As such, the policies that inform the development and operation of those sub-national IIS exist at the state and sometimes city levels.

The COVID-19 pandemic has reinforced the critical importance of robust data systems to ensure the availability of timely, consolidated, and up-to-date patient immunization data to guide public health decisions and support vaccination efforts. The pandemic also exposed that many jurisdictions have capacity constraints to utilize and expand their state IIS to respond to public health emergencies given current policy frameworks [3].

Since IIS are maintained separately by different states or jurisdictions, there is considerable variation between IIS policies (such as mandated reporting) as well as IIS functional components (inclusive of IIS functionalities and technical capabilities, such as reminders/recalls). This variation presents challenges to maintaining complete and high-quality patient data. However, the variability also provides an opportunity to understand the specific impact of different IIS policies and functionalities on vaccination rates, population health efforts, and provider reporting of vaccinations to the IIS.

To understand the available evidence in this area, we conducted a targeted literature review to identify key policy and technical components of IIS. The objectives of our review were to (1) identify the various documented functional and policy components of an IIS and (2) understand the real-world impact of the functional components and policies.

## 2. Materials and Methods

### 2.1. Search Strategy and Selection Procedure

We used PubMed to identify literature published between January 2012 and January 2022. All countries and methodologies, including commentaries, were included. Non-English publications were excluded. The search string used was *(“vaccin*”[MAJR] OR “Immuniz*”[MAJR]) AND (“information storage and retrieval/methods”[MeSH Major Topic] OR “information systems/standards”[MeSH Major Topic] OR “information systems/trends”[MeSH Terms] OR “Information Systems/organization and administration”[MAJR]).*

Reviewers screened abstracts and full texts, with at least two reviewers independently screening each. Due to the small number of articles retrieved, title screening was not applied as part of the review.

Reviewers applied inclusion and exclusion criteria to screen abstracts and full text articles. We included all articles that discussed or evaluated components that could be included in an IIS. This included IIS functional components that were evaluated as electronic health record (EHR) components, such as clinical decision support and patient reminders. Articles were excluded if they discussed strategies to increase vaccination rates unrelated to IIS, reviewed the one-time creation of vaccine databases or registries for a specific initiative, health system, or pilot program, or were specific to vaccine adverse event monitoring. Discrepancies between inclusion and exclusion decisions were resolved through discussion to reach a consensus among researchers.

### 2.2. Data Extraction

Three researchers independently extracted the following information: article characteristics, including title, authors, data, region, and study design; study population (e.g., pediatric versus adult); IIS policy and/or functional component, such as centralized reminder/recall and bidirectional interoperability; and, where applicable, outcomes or direct impact of IIS policies and functional components, such as vaccination rates and vaccine data quality. Researchers extracted statistical outcomes (e.g., results associated with point estimates and measures of significance) from observational and experimental articles. Only the original results of systematic literature reviews (SLRs) were extracted; any secondary research results referenced in the SLRs had their original sources cited. Extraction conflicts were resolved through review and consensus discussions.

### 2.3. Data Analysis

Following extraction, researchers used thematic analysis to identify and categorize IIS policies and functional components found throughout the published literature. A frequency analysis was conducted to understand the percent of extracted statistical outcomes that were significant for each identified policy or functionality theme. Non-statistical impacts and outcomes were evaluated qualitatively.

## 3. Results

### 3.1. Search Strategy Results

The search string resulted in a total of 86 articles. After an abstract and full text review, 39 articles were included in the final analysis. Figure 1 outlines the search process, application of exclusion criteria, and articles included in the final analysis.

### 3.2. Article Characteristics

The articles were heterogeneous with respect to study design, interventions, outcomes, and effect measures. The characteristics of the articles included in the review are listed in Table 1. The most frequent study design type was observational without controls (41%), followed by randomized controlled trials (23%). Of all 39 articles, 29 (74%) were from the US; within the US, Colorado had the most published literature compared to other states, represented in six articles (21% of US articles). Eight articles were from other countries, which include: Australia (1), Brazil (1), China (2), Iran (1), Kenya (1), Ghana (1), Nigeria (1), and Zimbabwe (1). Lastly, 25 (64%) of the included articles focused on pediatric populations; nine (23%) articles focused on both adults and children; and only four (10%) articles were limited to adults only.

### 3.3. Identified IIS Policies and Functional Components

A total of 17 distinct IIS policies and functional components were identified across the 39 articles (Table 2), with many articles discussing multiple policies or functional components (Figure 2). Across the 39 articles, 118 extractions identifying or evaluating IIS policies or functional components were included. Provider-based patient reminder/recall was the most frequently identified functional component with 43 (36.4%) extractions, followed by clinical decision support (n = 14, 11.9%) and IIS-based centralized reminder/recall (n = 13; 11.0%).

### 3.4. Impact of IIS Policies and Functionalities

Table 3 includes all IIS policies and functional components identified in the literature, including outcomes supported either through statistical evaluation or qualitative evidence. In some cases, outcomes used different terms across articles (e.g., vaccination rate versus immunization rate) to refer to the same concept; we grouped similar concepts together. Some articles used multiple measures to assess one outcome. For example, one article assessed the impact of patient reminder/recall on vaccination rates and assessed both receipt of a Tdap, meningococcal conjugate, or HPV vaccine in a 6-month period post-intervention and receipt of all vaccines (Tdap, meningococcal conjugate, and HPV vaccine) in a 6-month period post-intervention [13]. Additionally, multiple articles examined the same general vaccination outcome but for different vaccines. All outcome measures and definitions used are listed in Table 3. Outcomes with an asterisk (*) indicate the finding is from a randomized control trial (RCT).

Only 45 (38.1%) of all 118 extractions included an evaluation of statistical significance, while 33 (28.0%) extractions cited the impacts of policies and functional components on real-world outcomes but did not statistically evaluate them. In this case, non-statistical impacts were extracted from all sections of the articles, but most commonly the results and conclusions sections. For example, the impacts of clinical decision support were extracted from the evidence synthesis section of Patel et al.’s Economic Review of Immunization Information Systems to Increase Vaccination Rates: A Community Guide Systematic Review [13]. Table 3 includes a complete list of all the evaluated impacts of IIS policies and functional components identified across the articles. For articles with statistical evaluations of the impact of IIS policy or functional component use, we included the proportion of results that showed significant differences in real-world outcomes. We reported non-statistical, evaluated findings, or qualitative support, by article. 

## 4. Discussion

Out of the 17 IIS policies and functional components identified in our targeted literature review, patient reminder/recall was discussed the most in the published research and was associated with increased vaccination rates and vaccine knowledge [34,38]. This IIS functional component may be a key intervention to help increase vaccine uptake across the life course and can also provide opportunities for vaccine education and strengthening vaccination confidence, such as targeted reminder messages with information on the importance of vaccination [39]. However, the different approaches to generating reminder calls may have varied effectiveness. For example, two studies found that centralized reminder/recall interventions coming directly from IIS or Health Departments were more effective in increasing vaccination coverage and cost-effective compared to practice-based reminder/recall interventions—despite both being effective strategies to support vaccine uptake. This could be due to provider burden, capacity, and resources needed to implement practice-based reminders/recalls [19,20]. According to a 2019 Association for Immunization Managers (AIM) survey, only 20 out of the 53 immunization programs surveyed conducted centralized, IIS-based reminder/recall activities [40], providing an opportunity for additional states to explore utilizing this functional component.

Another key functional component with evidence of positive impact that could support vaccination coverage across the life course was clinical decision support. Clinical decision support systems aid providers in administering accurate and timely vaccinations in accordance with Advisory Committee on Immunization Practices (ACIP) recommendations. Four articles demonstrated that implementation of a clinical decision support functional component, such as forecasting and prompting recommended vaccinations for patients, reduced the administration of unnecessary vaccinations and increased vaccination rates [21,22,23,24]. However, according to an assessment conducted by the American Immunization Registry Association (AIRA) in 2021, only 51% of jurisdictional IIS included in the assessment met the twelve criteria necessary for clinical decision support [41].

Despite the lack of studies that lend themselves to statistical analysis in published literature, it is a well-established belief that immunization data interoperability is critical to supporting IIS data quality, information availability and access across provider types and sites of care, and cross-jurisdictional exchange of data [16,42,43]. A complete, accurate, and timely exchange of vaccination data between EHR and IIS supports other IIS functional components, such as reminder/recall and clinical decision support, that aid providers in administering timely and appropriate vaccinations. For instance, data exchange between EHRs and New York City’s IIS was associated with an increase in the percentage of children with up-to-date vaccination status and increased vaccination record completeness [44]. While 54 of 55 evaluated IIS by AIRA in the US can send consolidated patient immunization records to EHRs [45], barriers remain with regard to full national interoperability. In 2019, 51% of EHRs could not receive immunization record data from state IIS, and some required health systems to pay a fee to maintain a connection with the IIS [46]. Additionally, with an increasingly mobile society, interstate data sharing between jurisdictional IIS could be critical to ensuring complete patient data; however, limited data exists in the literature to fully demonstrate the value of cross-justification IIS data exchange.

We identified relatively few articles in our review evaluating the impact of different IIS policies and functional components on the adult population, as most articles focused on pediatric vaccinations and outcomes. The limited amount of research evaluating the outcomes of IIS policies and functional components is surprising, given that many state IIS have been operational since the 1990s [2]. This demonstrates a critical gap in the literature, which may be due, in part, to the difficulty in evaluating the impact of IIS policies and functional components on health outcomes. This demonstrates a critical gap in the literature, which may be due, in part, to the difficulty in evaluating the impact of IIS policies and functional components on health outcomes. National COVID-19 vaccination efforts have accelerated adult immunization capture in the US, with the number of adults with at least one vaccine documented in the IIS increasing from 68% to 89% between 2020 and 2021 [47]. As efforts are underway to increase adult vaccination coverage rates, more work is needed to evaluate the impact of IIS functional components and policies to inform state efforts to improve adult vaccination, including measures that can be implemented to improve adult data completeness in IIS. The number of Americans ages 65 and older is projected to double and become more racially and ethnically diverse between 2018 and 2060 [48]. Having complete patient data for these populations, including demographic data, is critical to ensuring vaccinators can understand patient vaccination needs (especially important for patients with chronic health conditions that are at increased risk for complications from certain vaccine-preventable diseases), as well as understand and address any demographic disparities that may exist within communities.

While the US continues to work toward optimizing the vaccine ecosystem, other countries offer useful insights into promising IIS practices and lessons learned. For instance, Australia has been recognized for its advanced IIS policies [12], which include lifetime patient enrollment, mandatory provider reporting, and patient access to vaccination data [49,50]. Canada also offers useful lessons since their IIS landscape and challenges are comparable to the US, such as multiple sub-national IIS managed by the different territories and provinces and considerable variation across policies and functional components despite national standards. For instance, Canada has faced challenges in the implementation of Panorama, deployed in 2011 and intended to be a national electronic information system to support public health surveillance, including immunization and disease outbreak management [51]. While significant effort was invested in national implementation, Panorama is not used by all provinces and territories, many of which have developed their own platforms with variations in functional components [52,53]. Canada is working to overcome similar challenges to the US to advance sub-national IIS interoperability and data sharing across all regions, such as through the consistent use of standardized codes for vaccine data [54], and may therefore offer useful insights.

Although the CDC maintains IIS functional standards and core data elements [8], they are not federally mandated, and there are challenges in prioritizing the standardization and optimization of state IIS policies and functional components. The lack of centralized requirements and connectivity provides states with flexibility for IIS policy design and management, and downstream, it leads to variation across IIS, posing barriers for cross-jurisdictional exchange of immunization data. For example, variation in state IIS policies for required race and ethnicity data reporting has impeded national efforts to identify and address vaccination disparities [2]. Inconsistent immunization data captured also limits understanding of the full extent of the pandemic’s impact on routine vaccination rates in the US. Recently, the CDC announced the deployment of a Data Use Agreement to jurisdictions in order to facilitate the submission of vaccination data as early as January 2023—aiming to help capture data across all IIS to support informed and timely public health decisions.

Funding from the CDC and other national initiatives are driving momentum to improve IIS infrastructure and broader data modernization, which provides a window of opportunity to implement evidence-based policies and technical improvements. The CDC received funding from the 2021 American Rescue Plan to support state and local-level COVID-19 vaccination efforts, including for health information technology and data system enhancements to facilitate immunization data sharing and vaccination efforts among underserved groups [2]. As funding support continues for state immunization programs to combat the COVID-19 pandemic, further evidence is needed to optimize the use of funds.

There are several limitations to our research. First, our search was restricted to articles in English, and most articles focused on the US as well as the pediatric population. Consequently, an in-depth evaluation between countries or age groups was not able to be performed, so results may not be generalizable to non-US countries or across all age groups pediatric populations. Second, to comprehensively review the limited available literature, we grouped results from articles with different levels of evidence, populations, and study designs and included editorials and opinion pieces, limiting the strength of our conclusions. Third, the majority of study designs used in identified articles did not lend themselves to statistical analysis of IIS impacts, limiting our ability to draw conclusions about the impacts (e.g., on vaccination rates) of many IIS policy and functional components, such as protection of health information and alternative site participation. We relied on article findings and did not independently validate the content within the articles captured in this review [55].

## 5. Conclusions

Through a targeted review of the literature, patient reminder/recall, centralized reminder/recall, and clinical support had the most published research to suggest their implementation improves real-world outcomes. However, additional research is needed to guide states toward prioritizing and implementing the most impactful IIS policies and functional components. Additional research can further demonstrate the value of robust IIS and support efforts to ensure sustainable IIS funding. There is a need to robustly assess the impact of policies and functional components to guide the development of best practices for IIS, support a more standardized and interoperable national IIS landscape, and ultimately improve vaccination coverage and population health.

## Figures and Tables

**Figure 1 vaccines-11-01242-f001:**
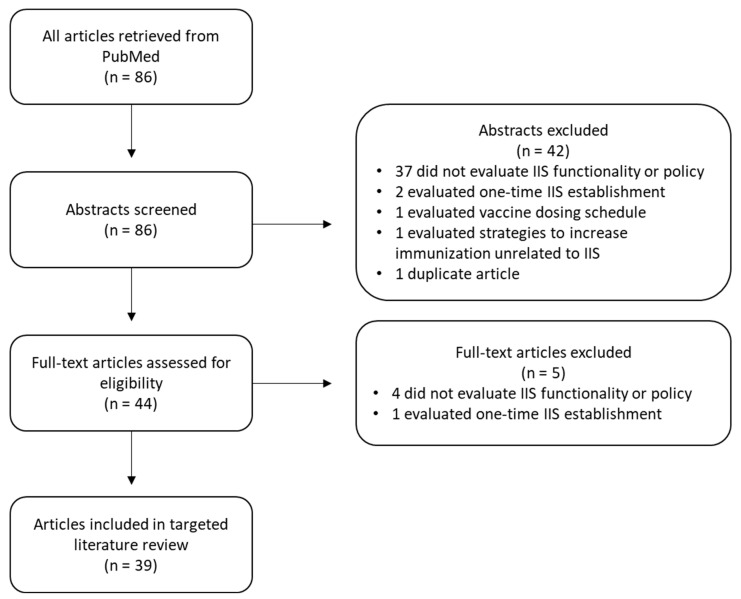
Published Literature Search and Screening Process.

**Figure 2 vaccines-11-01242-f002:**
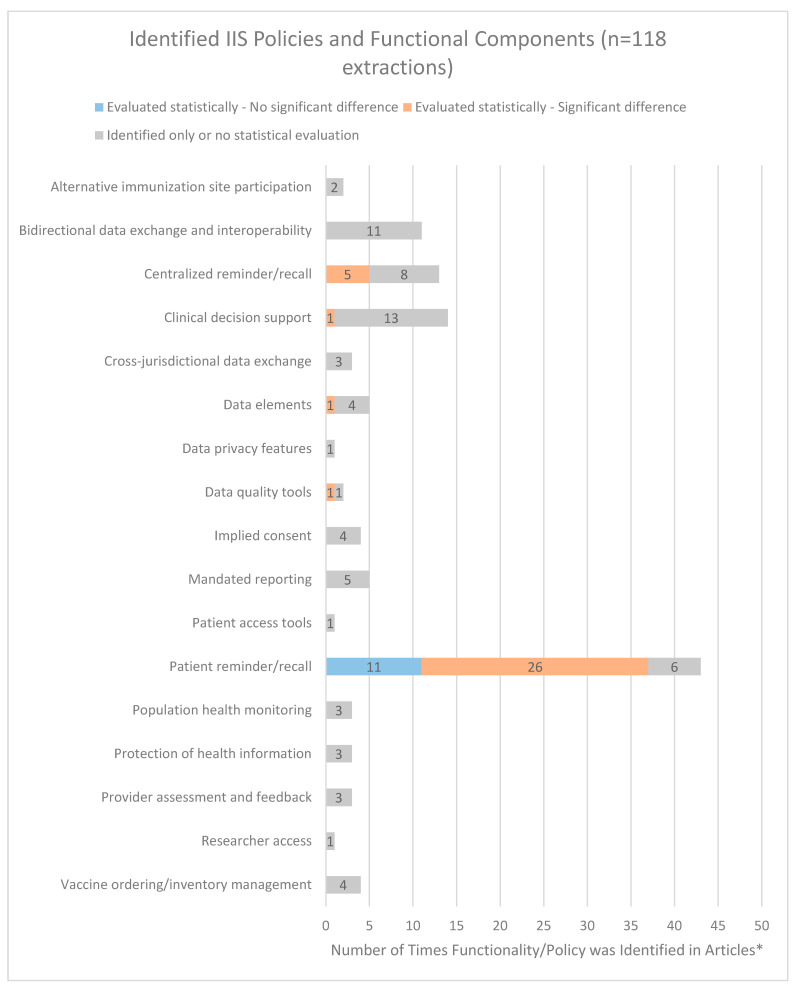
Frequency of IIS Policies and Functional Components Identified in Published Articles. * Each of the 39 articles could include multiple outcomes.

**Table 1 vaccines-11-01242-t001:** IIS Policy and Functionality Published Article Characteristics (n = 39).

Article Characteristic	Articles (n = 39)
**Study Design and Level of Evidence**	
Level 1: Randomized control trial	**9**
Level 2: Non-randomized trial	**1**
Level 3: Observational article with controls	**5**
Level 4: Observational article without controls	**16**
Other: Editorial, review article, etc.	**8**
**Geographic Area**	
Australia	**1**
Brazil	**1**
China	**2**
Iran	**1**
Kenya and Ghana	**1**
Nigeria	**1**
Zimbabwe	**1**
USA	**29**
Colorado (n = 5)	
Indiana (n = 1)	
Massachusetts (n = 1)	
Michigan (n = 1)	
Minnesota (n = 1)	
New York (n = 1)	
Texas (n = 1)	
Washington (n = 1)	
Multiple states (Colorado and New York) (n = 1)	
State not specified (n = 16)	
Geographic area not specified	**2**
**Adult or Pediatric Population**	
Pediatric only	**25**
Adult only	**4**
Pediatric and adult	**9**
Not specified	**1**

**Table 2 vaccines-11-01242-t002:** Identified IIS Policy and Functional Component Themes and Definitions.

IIS Policy/Functional Component Theme	Definition
Alternative site participation	IIS policies and functionalities that support alternative site access and use of IIS data, such as pharmacy, school, and childcare settings [4]
Bidirectional data exchange	Policies and functionalities that enable data systems to exchange data in two directions (e.g., two-way data exchange between a vaccine provider’s EHR and the IIS); this functionality allows providers to submit vaccination data to the IIS and query the IIS for patient records and vaccination forecast [5]
Centralized reminder/recall	Functionality that enables IIS- or Department of Health-based reminders to patients about upcoming scheduled or overdue vaccinations [6]
Clinical decision support	IIS functionality that automatically identifies the appropriate vaccinations for an individual and delivers this information to the vaccination provider (also referred to as evaluation and forecasting) [7]
Cross-jurisdictional data exchange	Electronic data exchange between an IIS and IIS in other jurisdictions consistent with interoperability standards; includes policy authorizing the request and receipt of immunization data from other IIS, the functionality of an IIS to query another IIS for immunization records, and the functionality enabling an IIS to send patient records to IIS in other jurisdictions for individuals located in those jurisdictions [4]
Data elements	Policy dictating the types of data that must or should be submitted to the IIS to record patient demographics and vaccination events; includes state-level requirements as well as CDC-endorsed data elements [4,8]
Data quality tools	Functionalities embedded in IIS that ensure that data is complete, accurate, consistent, and timely, including automated processes for identifying, preventing, and resolving fragmented patient records and duplicate vaccination events [4]
Implied consent	Policies where patients are by default included in IIS and must actively opt out of participation; alternatively, policies may stipulate patients cannot opt out of participation, this would be mandated participation though, not implied consent. [9]
Interoperability	The functionality of two systems (e.g., IIS and EHR) to exchange data electronically and use the exchanged data, supported by common transport protocols, messaging formats such as Health Level 7 (HL7), and code sets [10]
Mandated reporting	Policy requiring vaccinating providers or sites of care to submit immunization data to the IIS, which can be limited to specific age groups or vaccines [9]
Patient access tools	Functionality that allows patients to access their immunization record in the IIS (e.g., online patient IIS access portal) [11]
Patient reminder/recall	Functionality enabling provider- or practice-based reminders to patients about upcoming or overdue vaccinations (e.g., phone calls from providers to patients about upcoming vaccinations) [6]
Population health monitoring	IIS-based, population-level surveillance of vaccination rates and preventable disease rates or outbreaks, including population vaccination coverage assessments, support for public health emergency responses, and policies and functionalities supporting IIS access for public and population health purposes [4]
Protection of health information	The functionalities and policies that ensure that patient information is securely stored and shared, including confidentiality policies, IIS-enrolled site and user agreements, account management security policies (e.g., unique IIS log-in credentials), and physical and digital security protections and policies consistent with industry standards [4]
Provider assessment and feedback	Functionality enabling evaluation of and feedback on provider performance in delivering one or more vaccinations to a patient population [12]
Researcher access	Policies that allow researchers to request and access IIS data, including access to support vaccine-preventable disease investigation and control [4]
Vaccine ordering/inventory management	Functionalities that support vaccine ordering for provider sites, provide the status of vaccine orders, or organize and automatically decrement administered doses from vaccine inventory [4]

**Table 3 vaccines-11-01242-t003:** Summary of IIS Policy and Functional Components Findings in Published Literature.

IIS Functional Component or Policy	Outcome Category	Author Definition(s)	Summary of Findings [* Finding from RCT]
Alternative site participation	Data completeness	Percentage of state residents between 4 months and 5 years old with immunization data in the IIS	Qualitative support for increase [14]
Bidirectional data exchange and interoperability	Vaccine campaign effectiveness	Vaccine campaign effectiveness (no definition provided)	Qualitative support for increase [15]
	Data quality	Data quality (no definition provided)	Qualitative support for increase [16]
	Data completeness	Data completeness (no definition provided)	Qualitative support for increase [16]
Centralized reminder/recall	Vaccination series initiation rates	Initiation of the HPV 2- or 3-dose vaccine series	Significant increase in 1 out of 1 outcome [17] *Qualitative support for no impact [17] *
	Vaccination rate	Documentation of receipt of an influenza vaccine in IIS	Significant increase in 1 out of 1 outcome [18] *
		Percentage of patients with at least 1 documented immunization	Significant increase in 1 out of 1 outcome [19] *
	Up-to-date immunization status	The percentage of patients who achieved up-to-date immunization status	Significant increase in 1 out of 1 outcome [19] *
		The percentage of children with up-to-date vaccinations 6 months after recall	Significant increase in 1 out of 1 outcome [20] *
	Vaccine series completion or initiation	Completion of the HPV 2- or 3-dose vaccine series among adolescents aged 11 to 19 years	Qualitative support for no impact [17] *
	Cost of reminders compared to provider-based reminders	Cost of centralized reminder/recall using the state IIS compared to practice-based reminder/ recall	Qualitative support for decreased cost [19] *
		Cost of centralized reminder/recall using the state IIS compared to provider-based reminder/ recall	Qualitative support for decreased cost [20] *
Clinical decision support	Unnecessary vaccinations or overvaccination	The number of potentially unnecessary tetanus vaccine administrations in a baseline phase versus post-intervention	Significant decrease in 1 out of 1 outcomes [21]
		Over-immunization with a second influenza vaccine dose	Qualitative support for decrease [22]
		Overvaccination/vaccination duplication	Qualitative support for decrease [13]
	Immunization accuracy	Delivery of accurate childhood vaccinations (aligned with recommended and eligible vaccination timelines)	Qualitative support for increase [23]
	Vaccination rates	Immunization rates in adults 65 years of age or older and in younger adults with diabetes or chronic obstructive pulmonary disease	Qualitative support for increase [24]
	Administrative efficiency	Reduced administrative burdens associated with a vaccine delivery system	Qualitative support for increase [25]
	Immunization errors	Vaccination errors/adverse drug events reported through a voluntary system	Qualitative support for decrease [26]
	Vaccine-preventable outbreaks	Vaccine-preventable outbreaks (in general)	Qualitative support for decrease [27]
Data element requirements	Data timeliness	The number of days between immunization administration and submission of data to the IIS	Significant decrease in 1 out of 1 outcomes [28]
		Timeliness of IIS data exchange using HL7 connections	Qualitative support for increase [28]
	Data completeness	Data element completeness of IIS data exchanged through HL7 connections	Qualitative support for no impact [28]
		Completeness of data in IIS fields for vaccine manufacturer and lot number	Qualitative support for increase [29]
	Vaccination equity	Reduced gaps in vaccination coverage by race/ethnicity	Qualitative support for increase [15]
Data quality tools	Data consistency	Data consistency between routine immunization monthly summary forms and national health management information system summary forms	Significant increase in 1 out of 1 outcomes [30]
Mandated reporting	Data completeness	Completeness of IIS data	Qualitative support for increase [15]
	Patient participation in IIS	Adult participation in IIS	Qualitative support for increase [31]
Patient reminder/recall (provider- or EHR-based reminders)	Vaccination rate	Receipt of a tetanus-diphtheria-acellular pertussis (Tdap) vaccine, meningococcal conjugate, or the first dose of the human papillomavirus (HPV) vaccine; receipt of all targeted vaccines (Tdap, meningococcal conjugate, and the first dose of the HPV vaccine)	Significant increases in 2 out of 2 outcomes [32] *
		Receipt of a seasonal influenza vaccine among high-risk patient groups; receipt of a seasonal influenza vaccine among children younger than age five	Significant increases in 2 out of 3 outcomes [33] *
		Receipt of a meningococcal vaccine (MCV4); receipt of a pertussis vaccine (Tdap); receipt of an HPV-1 vaccine; receipt of an HPV-2 vaccine; receipt of an HPV-3 vaccine; receipt of all vaccinations (Tdap and all 3 HPV vaccines)	Significant increases in 7 out of 12 outcomes [34] *
		Receipt of scheduled vaccines at 6 weeks, 10 weeks, and 14 weeks	Significant increases in 3 out of 3 outcomes [35] *
		Vaccine uptake (multiple vaccine types)	Qualitative support for increase [36]
		Vaccination rates (in general)	Qualitative support for increase [37]
	Up-to-date immunization status	Receipt of MCV4, Tdap, and all three HPV vaccines among adolescent girls	Significant increases in 3 out of 4 outcomes [33] *
	Vaccine knowledge	Knowledge about the HPV vaccine	Significant increase in 1 out of 1 outcomes [38] *
	Vaccination appointment scheduled	Vaccination appointment scheduled	Significant increase in 1 out of 1 outcomes [39]
	Missed vaccination opportunities	Primary care visits during which a vaccine (MCV4, Tdap, and HPV vaccines) was due but not administered	Significant decreases in 4 out of 8 outcomes [34] *
	Delay of immunization receipt	Those who did not delay receiving scheduled immunizations at 14 weeks; median delay of receipt of scheduled immunizations at 14 weeks	Significant decreases in 2 out of 2 outcomes [35] *
	Time to vaccination	The average number of days between the start of the study period and when seasonal influenza vaccinations occurred	Significant decrease in 1 out of 1 outcomes [33] *
	Vaccine completion rate	Completion of the HPV 3-dose series	Qualitative support for no impact [38] *
	Revenue	Revenue generated by both standard and enhanced reminder/ recall (after subtracting the cost of reminder/ recall activities)	Qualitative support for increase [39]
	Vaccine-preventable diseases	Vaccine-preventable diseases (in general)	Qualitative support for decrease [37]
Protection of health information	Provider reporting	Vaccine provider reporting of immunization data to the IIS	Qualitative support for increase [15]
Provider assessment and feedback	Vaccination rates	Vaccination rates (in general)	Qualitative support for increase [37]
	Vaccine-preventable diseases	Vaccine-preventable diseases (in general)	Qualitative support for decrease [37]

## Data Availability

No new data were created or analyzed in this study. Data sharing is not applicable to this article.

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
