# Peer review of "Understanding Diversity of Policies, Functionalities, and Operationalization of Immunization Information Systems and Their Impact: A Targeted Review of the Literature"

_vaccines, 2023, doi:10.3390/vaccines11071242_

Round 1
Reviewer 1 Report
In this paper, Donckels et al performed a targeted literature review of immunization information systems (IIS), specifically to identify different IIS policies, the functionalities of IIS, and the impact of their use. They included 39 articles in the analysis, and found wide heterogeneity in terms of study design, interventions, outcomes, and effect measures. Overall, patient reminder/recall was the most commonly evaluated functional component (either provider-based or centralized), followed by clinical decision support. Patient reminders and recalls were associated with increased vaccine uptake and improved vaccine knowledge. However, the authors noted in their review that interoperability between existing IIS remains limited, and further efforts are needed to harmonize the capture and sharing of data about vaccine history. This is important for public health decision- and policy-makers, as it highlights gaps in evidence and practice regarding data capture of vaccination histories.
I read this paper with great interest, and I must say that it is the first time in a long time that I actually have no comments to make at all. The authors did a comprehensive (and reproducible) data search, and summarized all the findings clearly. The article is very well written, and all the important points are covered in the discussion.
Overall, the authors are to be commended for this excellent work.
(If one really wants to “nit pick”, then there is a comma on line 236 after reference [2] that should probably be a period. This could be corrected on the proofs.)
Author Response
Reviewer 1: (If one really wants to “nit pick”, then there is a comma on line 236 after reference [2] that should probably be a period. This could be corrected on the proofs.)
- Response: Many thanks for highlighting this oversight. We have since corrected.
Reviewer 2 Report
1. The article’s topic is very interesting, as it focuses on a subject which deeply permeates modern day healthcare.
2. It would be interesting to know whether the quoted United States’ IISs are regulated by state or country law, and to which extent.
3. One of the main findings of the paper is that only a few studies have been written regarding immunization information systems. This point should be highlighted in the article’s discussion, as it shows that still little attention is dedicated to information systems applied to healthcare.
4. Line 236: there is a small typo (comma used instead of a full stop). I suggest scanning the text for potential mistakes.
5. Discussion. Authors should reinforce the importance of strategy to improve vaccination coverage in chronic patients (considering the above mentioned articles):
· vaccination offer during hospitalization or outpatient follow up visit - doi: 10.1016/j.ajic.2017.10.004; doi: 10.3238/arztebl.2019.0645
· key role of mass media - doi: 10.1136/bmjgh-2020-004206, doi: 10.3389/fped.2022.869893
· transparency in safety and effectiveness data - doi: 10.1002/14651858.CD015477, doi: 10.1016/j.ajic.2021.10.015
A minor editing of English language is required
Author Response
- It would be interesting to know whether the quoted United States’ IISs are regulated by state or country law, and to which extent.
- Response: We thank the reviewer for this important question. We clarified in the introduction that the laws that govern IIS in the US are at the state and city level, since the quoted language is regulated by state/city law. The added language is pasted below in italics.
There are over 60 IIS in the United States (US) operating at the sub-national level in all fifty states, Washington, D.C., and select cities and territories [2]. As such the policies that inform the development and operation of those sub-national IIS exist at the state and sometimes city level.
- One of the main findings of the paper is that only a few studies have been written regarding immunization information systems. This point should be highlighted in the article’s discussion, as it shows that still little attention is dedicated to information systems applied to healthcare.
- Response: We thank the reviewer for identifying this. We added a line to underscore this in the discussion. The added language is pasted below in italics.
We identified relatively few articles in our review evaluating the impact of different IIS policies and functional components among the adult population, as most articles focused on pediatric vaccinations and outcomes. The limited amount of research evaluating the outcomes of IIS policies and functional components is surprising, given that many state IIS have been operational since the 1990s [2]. This demonstrates a critical gap in the literature which may be due, in part, to the difficulty in evaluating the impact of IIS policies and functional components on health outcomes.
- Line 236: there is a small typo (comma used instead of a full stop). I suggest scanning the text for potential mistakes.
- Response: Many thanks for highlighting this oversight. We have since corrected.
- Discussion. Authors should reinforce the importance of strategy to improve vaccination coverage in chronic patients (considering the above mentioned articles):
- Response: We thank the reviewer for this suggestion. We have added emphasis on the importance of having complete vaccination data to improve vaccination coverage for patients with chronic health conditions in the discussion. The added language is pasted below in italics.
Having complete patient data for these populations, including demographic data, is critical to ensuring vaccinators can understand patient vaccination needs (especially important for patients with chronic health conditions that are at increased risk for complications from certain vaccine-preventable diseases), as well as understand and address any demographic disparities that may exist within communities.
Reviewer 3 Report
While the idea of this research is novel, it is not unclear how what is reported relates to the title.
Why should authors extract IIS information from articles - why not get list of IIS databases from a public source?
Authors should be clear they're discussing content about IIS extracted from articles published about IIS.
The methodology section should be sharper - content analysis, text analytics, statistical methods, etc.
Scope & Limitations must be discussed.
Literature review is insufficient.
English is ok.
Author Response
- While the idea of this research is novel, it is not unclear how what is reported relates to the title. Why should authors extract IIS information from articles - why not get list of IIS databases from a public source?
- Response: We appreciate the question. In this research, we were interested in first understanding the landscape of literature and existing evidence as it relates to IIS. This was a necessary step in order to understand what aspects of IIS have or have not been studied. As we mention in the introduction, in the US alone, there are over 60 different IIS that are operationalized at the subnational level. There is no publicly available, comprehensive, and up to date repository of state-by-state IIS functionalities and policies in the US. Furthermore, this literature review went beyond compiling information from IIS databases to also assess how specific IIS characteristics impact immunization programs. There have been great strides made in enhancing IIS over the last decade and to the authors’ knowledge, no literature review has been completed. Given this, our first step was to understand the current landscape from an evidence review perspective in order to identify opportunities for further research.
- Authors should be clear they're discussing content about IIS extracted from articles published about IIS.
- Response: In our methods we begin line 62 stating that we are identifying literature published between January 2012 and January 2022, and specify the search string that we used. As we state, starting in line 73, “We included all articles that discussed or evaluated components that could be included in an IIS. This included IIS functional components that were evaluated as an electronic health record (EHR) component, such as clinical decision support and patient reminders. Articles were excluded if they discussed strategies to increase vaccination rates unrelated to IIS, reviewed one-time creation of vaccine databases or registries for a specific initiative, health system, pilot program, or were specific to vaccine adverse events monitoring”
It would be helpful if reviewer 3 could provide more information on what we may be lacking in clarity here and would be happy to make any additional changes.
- The methodology section should be sharper - content analysis, text analytics, statistical methods, etc.
- Response: We appreciate Reviewer 3’s comment here. As this was a targeted literature review, there were no text analytics or statistical methods applied. We provide counts of unique articles and outcomes, plus integration of articles quantitative results (i.e., interpreting significance of odds rations and p-values). We followed similar methodology for other targeted literature reviews. Should Reviewer 3 have specific suggestions, we would be happy to explore additional edits.
- Scope & Limitations must be discussed.
- Response: We discuss scope of the literature review in the methodology section. Starting line 62 we provide the following: We used PubMed to identify literature published between January 2012 to January 2022. All countries and methodologies, including commentaries, were included. Non-English publications were excluded. The search string used was (“vaccin*”[MAJR] OR “Immuniz*”[MAJR]) AND (“information storage and retrieval/methods”[MeSH Major Topic] OR “information systems/standards”[MeSH Major Topic] OR “information systems/trends”[MeSH Terms] OR “Information Systems/organization and administration"[MAJR]).
Following this, starting lines 72-79, speak to the inclusion and exclusion criteria that were applied to the literature review. Should there be additional details Review 3 thinks would be important to include as it pertains to scope, we certainly welcome that and would make any edits.
Limitations section begins on line 254 through 262 (pasted below for reference). Should we need to provide more clarity on the limitations of this paper, please do let us know and we can explore additional edits. We did add one sentence in response to this comment that I have included in bold below.
There are several limitations to our research. First, our search was restricted to articles in English and most articles focused on the US. Second, to comprehensively review the limited available literature, we grouped results from articles with different levels of evidence, populations, and study designs and included editorials and opinion pieces, limiting the strength of our conclusions. Third, the majority of study designs used in identified articles did not lend themselves to statistical analysis of IIS impacts, limiting our ability to draw conclusions about the impacts (e.g., on vaccination rates) of many IIS policy and functional components, such as protection of health information and alternative site participation. We relied on article findings and did not independently validate the content within articles captured in this review.
- Literature review is insufficient.
- Response: It is unclear what reviewer 3 means here for us to be able to make additional changes. This was a targeted review of the literature as it relates to IIS to collate existing studies and subsequent evidence around IIS. The purpose of this was to identify policy and technical components associated with improved vaccine uptake, identify gaps in the literature, and provide decision makers with a comprehensive synthesis of the existing data.
Reviewer 4 Report
This is a targeted literature review on immunization information system (IIS), which helpful to understand the important element of IIS. The article is well described to follow the standard procedure of literature review. I would have an in-depth discussion on the country’s differences and
1) Nine articles reported the randomized control trial, as shown in Table 1. Flag the findings from the randomized control trial in Table 3. This would allow readers identification of the study quality.
2) 29 out of 39 reports focused on the IIS in the USA, while three articles were in African countries. Discuss the difference in the outcome measures between the countries. I expect that there will be a different status in the modernization and functionality of IIS.
3) Similarity, 25 out of 39 reports focused on the IIS for the pediatric population. Discuss the difference in the outcome measures between the age-defined populations.
Author Response
1) Nine articles reported the randomized control trial, as shown in Table 1. Flag the findings from the randomized control trial in Table 3. This would allow readers identification of the study quality.
- Response: We have added asterisks indicating which results in Table 3 are from an RCT.
2) 29 out of 39 reports focused on the IIS in the USA, while three articles were in African countries. Discuss the difference in the outcome measures between the countries. I expect that there will be a different status in the modernization and functionality of IIS.
- Response: We agree with your observation—see analysis below. However, because there were so few ex-US studies, we did not do an in-depth analysis to examine country level details. We did add this as a limitation.
3) Similarity, 25 out of 39 reports focused on the IIS for the pediatric population. Discuss the difference in the outcome measures between the age-defined populations.
- Response: Similar to above, we did not complete an evaluation between adult and pediatric outcomes; however, we do make note of this in the limitations.
Round 2
Reviewer 3 Report
Minor Revision 1. Still the title doesn't match research. Are established IIS policies searched for in the review papers or IIS policies are extracted from the review papers? 2. Literature review:Authors may add additional references with overall research theme, methodology; not papers collected for review itself.
Author Response
- Minor Revision 1. Still the title doesn't match research. Are established IIS policies searched for in the review papers or IIS policies are extracted from the review papers?
- Response: We appreciate the question. In this research, we were interested in first understanding the landscape of literature and existing evidence as it relates to IIS. As such we are amending the title accordingly.
“Understanding Diversity of Policies, Functionalities, and Operationalization of Immunization Information Systems and their Impact: A Targeted Review of the Literature”
- Literature review: Authors may add additional references with overall research theme, methodology; not papers collected for review itself.
- Response: We thank reviewer 2 for this comment; however we are unclear what the reviewer is asking us to do.